# UV/TiO₂ Photocatalysis as an Efficient Livestock Wastewater Quaternary Treatment for Antibiotics Removal

**Yeji Park [1], Sanghyeon Kim [2], Jungyeon Kim [2], Sanaullah Khan [3,\*] and Changseok Han [1,2,\*]**

[1] Department of Environmental Engineering, INHA University, Incheon 22212, Korea; yejipark@inha.edu
[2] Program in Environmental & Polymer Engineering, Graduate School of INHA University, INHA University, Incheon 22212, Korea; 22211303@inha.edu (S.K.); 22221388@inha.edu (J.K.)
[3] Department of Chemistry, Women University, Swabi 23430, Pakistan
[\*] Correspondence: sukhan3@gmail.com (S.K.); hanck@inha.ac.kr (C.H.)

**Abstract:** Antibiotics are the most common pharmaceutical compounds, and they have been extensively used for the prevention and treatment of bacterial diseases for more than 50 years. However, merely a small fraction of antibiotics is metabolized in the body, while the rest is discharged into the environment through excretion, which can cause potential ecological problems and human health risks. In this study, the elimination of seventeen antibiotics from real livestock wastewater effluents was investigated by UV/TiO₂ advanced oxidation process. The effect of process parameters, such as TiO₂ loadings, solution pHs, and antibiotic concentrations, on the efficiency of the UV/TiO₂ process was assessed. The degradation efficiency was affected by the solution pH, and higher removal efficiency was observed at pH 5.8 and 9.9, while the catalyst loading had no significant effect on the degradation efficiency at these experimental conditions. UV photolysis showed a good removal efficiency of the antibiotics. However, the highest removal efficiency was shown by the UV/photocatalyst system due to their synergistic effects. The results showed that more than 90% of antibiotics were removed by UV/TiO₂ system during the 60 min illumination, while the corresponding TOC and COD removal was only 10 and 13%, respectively. The results of the current study indicated that UV/TiO₂ advanced oxidation process is a promising method for the elimination of various types of antibiotics from real livestock wastewater effluents.

**Keywords:** livestock wastewater treatment; antibiotics; advanced oxidation processes; UV/TiO₂ photocatalysis



## 1. Introduction

The huge increase in the human population in the last several decades in the world has increased the demand for more food resources, which has led to rapid growth in the livestock industry [1–3]. However, because livestock animals were likely prone to bacterial infections, it has led to increased antibiotics usage for preventing bacterial infections and promoting the sustainable growth of livestock animals [4–7]. Approximately 100,000–200,000 tons of antibiotics were annually used all over the world [8]. Furthermore, the production of antibiotics has greatly increased in recent years. For instance, in China alone, the annual antibiotics usage increased by more than 10 times over a decade, i.e., rising from 6000 tons to 78,200 tons during 2003–2013 [9,10]. It is expected that antibiotics consumption will further increase by two-thirds (i.e., 105,600 tons) by 2030. Another study reported that antibiotic consumption from 2017 to 2030 in 41 countries was estimated to be 104,079 tons by 2030 [7]. These situations suggest that antibiotics can be consistently detected in the environment [11]. Livestock wastewater effluents represent one of the most common sources of antibiotics in the environment [12,13]. The concentration of antibiotics detected in the environment varied from ppt (ng/L) to ppb (μg/L) level [14–16], but the eco-toxicity was detected at ppm level (mg/L) [17,18].

The physicochemical properties of antibiotics include high molecular weight compounds, various structures and bioavailability depending on hydrophobicity. Because of this, they do not decompose completely in the human or animal body, influencing their transport in the environment [19]. Due to their persistent and non-biodegradable nature, antibiotics may likely accumulate in aquatic systems, plants, and animals [9]. Photocatalyst resistant bacteria may be introduced into the environment, which can directly or indirectly invade the human body [20]. In addition, only a small fraction of antibiotics is absorbed or metabolized in the body, while the rest are released into the environment through excretion [21]. Anthony et al. reported that 10–20% of the antibiotics were metabolized in the body [9], or an average metabolic rate may only be 30% [20]. It was also reported that 25–75% of the taken antibiotics were excreted from the bodies [22]. Antibiotics are non-biodegradable in the environment [23]. Owing to their polar structure, antibiotics are not absorbed on the subsoil [24] and hence persist in the environment for a long time [25]. Consequently, the widespread use of various antibiotics caused different adverse effects in humans, including dermatitis, gastrointestinal symptoms, carcinogenicity, reproductive effects, and teratogenicity [26–28]. The Center for Disease Control and Prevention (CDC) reported that antimicrobial-resistant bacteria caused diseases in more than 2,000,000 people in the US, and about 700,000 deaths were annually caused due to antimicrobial-resistant bacteria all over the world [29]. Therefore, it is of great importance to apply reliable treatment technologies to completely remove antibiotics from livestock wastewater.

Different wastewater treatment processes, including biological and physicochemical methods (i.e., coagulation, adsorption, membrane separation etc.), have been long employed for treating livestock wastewater effluents [2,30–38]. Although these treatment processes were usually economic, major disadvantages included longer reaction time (i.e., month) by the biological methods [2]. Some physicochemical treatment processes demonstrated drawbacks, including low efficiency and high operating expenses in terms of plant construction and disposal of the produced sludge [35]. The conventional biological and physicochemical methods can only partially destroy antibiotics [38]. In the case of South Korea, Kim et al. researched the characteristics of influent and effluent wastewater, including four types of antibiotics of monitored concentration with five types of treatment plants such as sequencing batch reactor, liquid-phase flotation, membrane bioreactor, bioreactor plus ultrafiltration (BIOSUF) and bio best bacillus systems. Although the result of this study was obtained removal efficiency of at least 90%, some of them such as chlortetracycline (483.7 μg/L→11.5 μg/L), sulfamethazine (251.2 μg/L→20.8 μg/L) and sulfathiazole (230.8 μg/L→28.2 μg/L) were not completely removed [11]. Therefore, new treatment methods are essentially required to develop reliable technologies for the complete removal of residual antibiotics after the biological process.

Advanced oxidation processes (AOPs) are promising alternatives to conventional wastewater treatment processes that can readily decompose antibiotic molecules or improve their biodegradability [37]. AOPs, such as ozonation, $UV/H_2O_2$, Fenton's reaction, electrochemical processes, photocatalysis etc., have been extensively used in many environmental remediations, especially for treating refractory organic pollutants [39–43]. The AOPs are usually characterized by the generation of reactive oxygen species, such as hydroxyl radicals ($^\bullet OH$) [44], which can attack and transform complex organic compounds, including antibiotics, into eco-friendly end-products, i.e., $CO_2$, $H_2O$ etc. [45,46]. The AOPs have shown fast reactivity and high efficiency for the degradation of antibiotics [26,47]. Ozone oxidation has been used for disinfection and oxidation of organic pollutants, and the decomposition rate increases as the pH increases [48]. However, as the ozonation process progresses, the pH decreases, and the generation of $^\bullet OH$ decreases, thereby decreasing the process efficiency [37]. Fenton's oxidation is a metal-catalyzed oxidation reaction in which iron is the catalyst [49], and the main disadvantage is the low pH requirement in order to prevent iron precipitation [34]. The electrochemical process had the disadvantage of a high operation cost [40].

Among the AOPs, TiO$_2$ photocatalysis has gained more attention for the decomposition of recalcitrant organic pollutants in the environment [22]. The TiO$_2$ photocatalyst can be activated by light with energy higher than the band gap energy of TiO$_2$, i.e., 3.2 eV. When TiO$_2$ was irradiated with ultraviolet (UV) light, different reactive oxygen species were generated through reactions of electron (e$^-$)-hole (h$^+$) pair, H$_2$O, and O$_2$ [44]. As a photocatalyst, TiO$_2$ is an inexpensive and non-toxic material [17,50]. Moreover, TiO$_2$ is photochemically stable with no mass transfer restrictions and a commercially approved material [51] that does not generate secondary pollutants [37]. However, it is difficult to remove and regenerate the photocatalyst [17], and several studies have solved the problem by combining other materials such as graphene or LDH with TiO$_2$. There were many studies in the literature about UV/TiO$_2$ photocatalytic degradation of individual antibiotics in water systems [40,48]. However, simultaneous degradation of an extended number of antibiotics (mixture of antibiotics) by UV/TiO$_2$ system, especially in the real livestock wastewater effluents, has not been reported so far.

In this study, the degradation of seventeen antibiotics (i.e., ceftiofur, clopidol, enrofloxacin, erythromycin, florfenicol, lincomycin, oxytetracycline, penicillin-G, penicillin-V, sulfadiazine, sulfamethazine, sulfamethoxazole, sulfathiazole, tetracycline, tiamulin, trimethoprim, and tylosin) in the real livestock wastewater effluents (i.e., Jeollanam-do, Korea), by UV/TiO$_2$ system was investigated. The effects of water quality and process parameters, such as TiO$_2$ loadings, solution pHs, and antibiotic concentrations, on the efficiency of the UV/TiO$_2$ system were assessed. The degradation efficiency of antibiotics by only UV or photocatalyst alone was also investigated for comparison purposes. Moreover, mineralization of the antibiotics in livestock wastewater was determined by measuring the reduction in UV$_{254}$ absorbance, and COD as well as TOC removal. Although factory-scale testing must also be performed prior to application at the industrial level, this study introduces lab-scale experiments. Since we tried to thoroughly understand aspects of photocatalysis in the treatment of real livestock wastewater effluents, all seventeen antibiotics together were spiked in the wastewater effluent. Moreover, UV/TiO$_2$ photocatalysis as a quaternary treatment process was applied for removing antibiotics in livestock wastewater effluent for the first time. The results of the study are expected to provide useful scientific information on the elimination of antibiotics from real livestock wastewater effluents by using an environmentally friendly UV/TiO$_2$ system.

## 2. Materials and Methods

### 2.1. Materials

The antibiotics used in this study, i.e., ceftiofur, clopidol, enrofloxacin, erythromycin, florfenicol, lincomycin, oxytetracycline, penicillin-G, penicillin-V, sulfadiazine, sulfamethazine, sulfamethoxazole, sulfathiazole, tetracycline, tiamulin, trimethoprim, and tylosin, were purchased from Sigma-Aldrich (St. Louis, MO, USA). The chemical formula and molecular structure of the antibiotics are presented in Table 1. The wastewater was collected from a real livestock wastewater treatment plant in Jeollanam-do, Korea. The characteristics of the wastewater are described in Table 2. The effluent was stored in a refrigerator prior to the degradation experiments or analysis. The TiO$_2$ used as a photocatalyst was Aeroxide P25. The pH of the reaction solution was adjusted by using 1 M NaOH or 1 M HCl solution. All chemicals were used as received.

**Table 1.** Chemical formula and molecular structure of the antibiotics.

| Antibiotics Name | Chemical | Structure |
|---|---|---|
| Ceftiofur | $C_{19}H_{17}N_5O_7S_3$ |  |
| Clopidol | $C_7H_7Cl_2NO$ |  |
| Enrofloxacin | $C_{19}H_{22}FN_3O_3$ |  |
| Erythromycin | $C_{37}H_{67}NO_{13}$ |  |
| Florfenicol | $C_{12}H_{14}Cl_2FNO_4S$ |  |
| Lincomycin | $C_{18}H_{34}N_2O_6S$ |  |
| Oxytetracycline | $C_{22}H_{24}N_2O_9$ |  |
| Penicillin-G | $C_{16}H_{18}N_2O_4S$ |  |
| Penicillin-V | $C_{16}H_{18}N_2O_5S$ |  |

**Table 1.** *Cont.*

| Antibiotics Name | Chemical | Structure |
| --- | --- | --- |
| Sulfadiazine | $C_{10}H_{10}N_4O_2S$ |  |
| Sulfamethazine | $C_{12}H_{14}N_4O_2S$ |  |
| Sulfamethoxazole | $C_{10}H_{11}N_3O_3S$ |  |
| Sulfathiazole | $C_9H_9N_3O_2S_2$ |  |
| Tetracycline | $C_{22}H_{24}N_2O_8$ |  |
| Tiamulin | $C_{28}H_{47}NO_4S$ |  |
| Trimethoprim | $C_{14}H_{18}N_4O_3$ |  |
| Tylosin | $C_{46}H_{77}NO_{17}$ |  |

**Table 2.** Characteristics of real livestock wastewater, obtained from a livestock wastewater treatment plant in Jeollanam-do, Korea.

| pH | Conductivity (mS/cm) | Total Organic Carbon (mg/L) | Chemical Oxygen Demand (mg/L) | UV$_{254}$ (cm$^{-1}$) | Color (mg/L Pt-Co) | Total Phosphorus (mg/L) | Total Nitrogen (mg/L) |
|---|---|---|---|---|---|---|---|
| $7.1 \pm 0.1$ | $778 \pm 3$ | $69 \pm 1$ | $262 \pm 2$ | $0.351 \pm 0.002$ | $30 \pm 2$ | $0.05 \pm 0.01$ | $5 \pm 1$ |

### 2.2. Analytical Methods

The degradation analysis of the antibiotics was conducted by using Liquid Chromatography Mass Spectrometry (LC-MS, Agilent 6460, Agilent, Santa Clara, CA, USA) and UV-Vis spectrophotometer (Thermo Fisher ScientificG10S, Thermo, Waltham, MA, USA), described in our previous paper [52].

The total organic carbon (TOC) was measured using a TOC-L$_{CPH/CPN}$ analyzer (SHIMADZU, Tokyo, Japan). Chemical oxygen demand (COD) was analyzed by COD$_{Cr}$ analysis [17].

### 2.3. Characterization of TiO$_2$ Photocatalyst

The morphological characterization of TiO$_2$ photocatalyst was investigated by using a transition electron microscope (TEM, JEM2100F, JEOL, Tokyo, Japan) and a scanning electron microscope (SEM, S-4300 SE, Hitachi, Tokyo, Japan). The surface of TiO$_2$ was analyzed by Fourier Transform Infrared Vacuum Spectrometer (FTIR, VERTEX 80V, Bruker, Karlsruhe, Germany) in the wavenumber ranging from 4000 cm$^{-1}$ to 400 cm$^{-1}$ with resolution of 4 cm$^{-1}$. The X-ray Diffractometer patterns of TiO$_2$ were measured by using Multi-purpose X-ray Diffractometer (X'pert Pro MPD, PANalytical, Almelo, Netherland) in the 2θ from 10° to 70° utilizing Cu, K$_\alpha$ radiation (λ = 1.54).

### 2.4. Photocatalytic Degradation Experiments

The photocatalytic experiments were performed in a batch mode photoreactor, consisting of a 500 mL quartz beaker. Figure 1 shows a schematic of experimental setups for this study. For UV irradiation conditions, two combined UV lamps (254 nm, 20 W, intensity = $16.8 \pm 1.5$ mW/cm$^2$) were located on both sides of 500 mL beaker. A quartz beaker was used since it is transparent to UV light (λ < 254 nm) rather than soda-lime glass which absorb it [53].

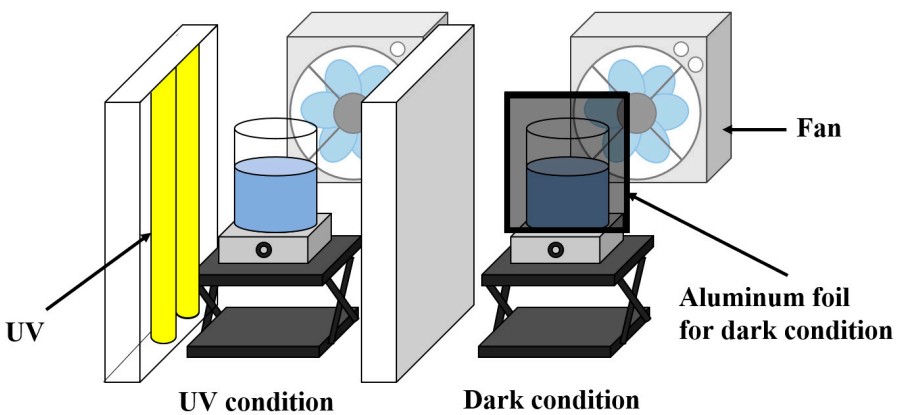

**Figure 1.** A Schematic of an experimental setup for photocatalytic antibiotics degradation.

In the dark experiments, the quartz beaker was covered with aluminum foil to block all light. The reactors were cooled with a fan during the experiments to avoid the effect of temperature on the removal of antibiotics. The reaction solution was kept under vigorous

stirring using a magnetic stirrer for maintaining homogeneity. To prevent further degradation or any adsorption effect on the sample during sample storage after the sampling, all taken samples were filtered using syringe filters (ADVANTEC® DISMIC®-13CP, 0.20 μm pore size, Tokyo, Japan), kept in glass vials, and stored in a refrigerator until sample analysis. All experiments were performed in duplicate.

## 3. Results and Discussion

### 3.1. Characteristics of TiO$_2$ Photocatalyst

As shown in Figure 2a, the TiO$_2$ nanoparticles used in this study were spherical and approximately 20 nm in diameter. XRD analysis in Figure 2b shows the presence of both phases of anatase and rutile in the TiO$_2$-P25 sample. Both phases of anatase and rutile in the sample were confirmed with the measured lattice spacing of 0.35 nm and 0.32, corresponding to the (101) plane of anatase and the (110) plane of rutile, respectively (an insert of Figure 2c). The specific surface area of TiO$_2$ (AEROXIDE P25) was 35~65 m$^2$/g. When UV light irradiates energy greater than the band gap (3.2 eV, [52]) of TiO$_2$ photocatalyst, free electrons (e$^-$) in the valence band (VB) are transferred to the conduction band (CB) of TiO$_2$, and thus, electron holes (h$^+$) formed in the valence band of TiO$_2$ [54]. The reaction mechanism of TiO$_2$ is shown through reactions 1–4 [55] and Figure 3. Reaction 2 shows a reduction process, while reactions 3 and 4 show an oxidation process. The pairs of e$^-$ and h$^+$ generated in CB and VB, respectively, produce oxygen-reactive species $^•$OH and O$_2^-$, and can remove pollutant compounds by redox reaction on the surface of TiO$_2$. Characterization of TiO$_2$ was also analyzed by FTIR. As shown in Figure 2d, FTIR analysis peaks appeared in the range 400–800 cm$^{-1}$, 1500–1700 cm$^{-1}$, and 2800–3600 cm$^{-1}$, and the main peak was observed at 400–800 cm$^{-1}$. According to reports, the peaks corresponding to the vibration of Ti-O are confirmed to be in the range 653–550 cm$^{-1}$ [56], while those related to Ti-O stretching and Ti-O-Ti bridging stretching modes are in the range 400–700 cm$^{-1}$ [57]. Maulidiyah et al. also confirmed the presence of peaks of TiO$_2$-P25 at 516 cm$^{-1}$–677 cm$^{-1}$ [58].

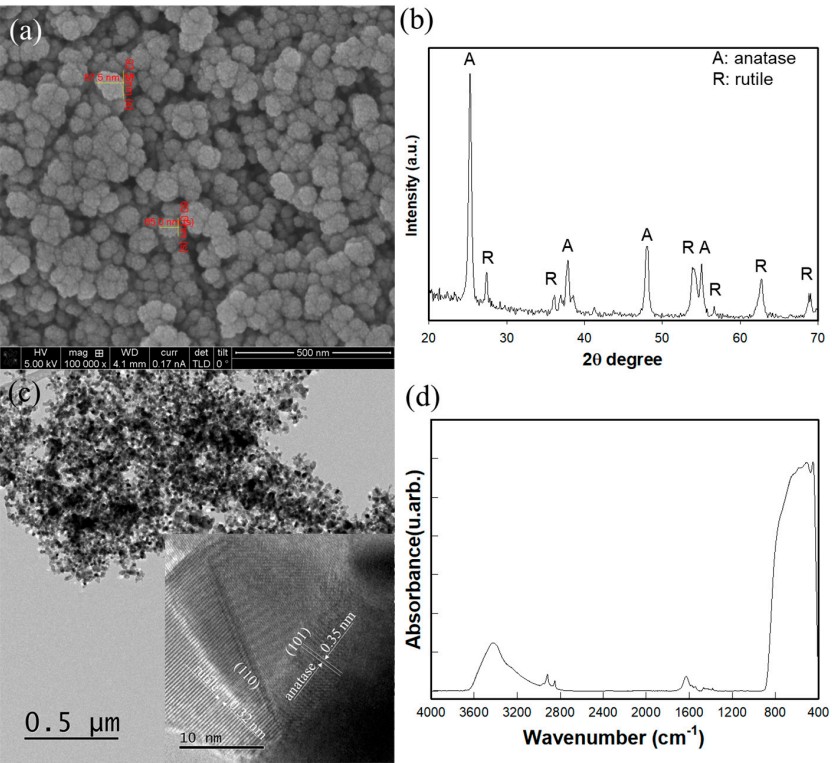

**Figure 2.** Characterization of TiO$_2$: (**a**) SEM analysis, (**b**) XRD analysis, and (**c**) TEM analysis (insert: HR-TEM analysis) and (**d**) FTIR analysis.

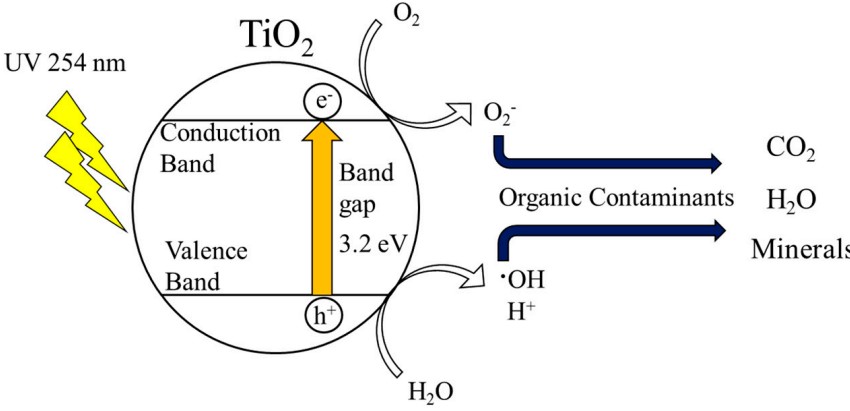

**Figure 3.** Schematic diagram of the photolysis Mechanism of $TiO_2$.

$$TiO_2 + h\nu \rightarrow TiO_2\ (e^-) + TiO_2\ (h^+) \tag{1}$$

$$TiO_2\ (e^-) + O_2 \rightarrow TiO_2 + O_2{}^- \tag{2}$$

$$TiO_2\ (h^+) + H_2O \rightarrow TiO_2 + {}^\bullet OH + H^+ \tag{3}$$

$$TiO_2\ (h^+) + HO^- \rightarrow TiO_2 + {}^\bullet OH \tag{4}$$

*3.2. UV/TiO$_2$ Photocatalytic Degradation of Antibiotics in Livestock Wastewater*

The degradation of antibiotics in livestock wastewater by UV/TiO$_2$ photocatalysis was investigated, and the results are shown in Figure 4. As can be seen, most of the studied antibiotics ([antibiotics]$_0$ = 20–100 µg/L, [TiO$_2$]$_0$ = 0.1 g/L) were effectively decomposed by TiO$_2$ photocatalysis under UV illumination for 30 min. The degradation of antibiotics in water by UV/TiO$_2$ process was mainly due to ${}^\bullet$OH produced by UV/TiO$_2$ system (reactions 1–4) [59] and/or direct UV photolysis [60]. The results shown in Figure 4 revealed that the seventeen antibiotics behaved differently towards the UV/TiO$_2$ process, as indicated from their removal efficiencies using the same reaction conditions. For instance, nine antibiotics, including ceftiofur, enrofloxacin, oxytetracycline, penicillin-G, penicillin-V, sulfadiazine, sulfamethazine, sulfathiazole, and tetracycline, were 100% degraded for 30 min of UV illumination. The degradation efficiency of five antibiotics, i.e., clopidol, lincomycin, tiamulin, trimethoprim, and tylosin, by the UV/TiO$_2$ system was around 90%. On the other hand, the remaining three antibiotics, i.e., erythromycin, florfenicol and sulfamethoxazole, showed less than 70% removal efficiency after 30 min of the photocatalysis. The difference in the removal efficiency of the tested antibiotics by the UV/TiO$_2$ process can be explained on the basis of structural differences [61]. Karaolia et al. showed the solar-light/TiO$_2$ photocatalytic degradation efficiency varied among the antibiotics, represented by 87, 10 and 19% removal of sulfamethoxazole, erythromycin, and clarithromycin, respectively, after 60 min illuminations [62]. Elmolla et al. showed that the degradation efficiency of amoxicillin, ampicillin and cloxacillin by UV/ZnO photocatalysis was 44, 60 and 96%, respectively, after 5 h, using 0.2 g/L ZnO [63]. Elmolla and Chaudhuri [51] reported the degradation rate constant (k) by UV/TiO$_2$ photocatalysis varied among the antibiotics, represented by 0.007, 0.003 and 0.029 min$^{-1}$ for amoxicillin, ampicillin and cloxacillin, respectively.

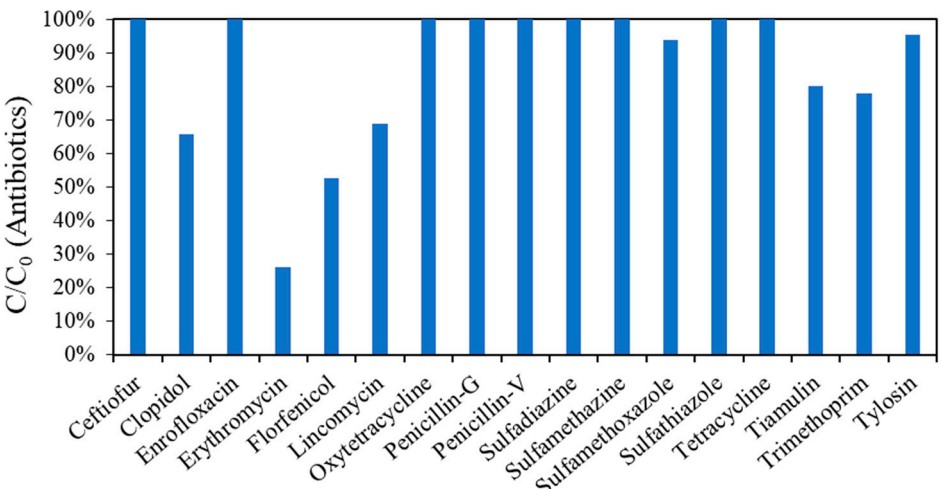

**Figure 4.** Photocatalytic degradation of antibiotics by UV/TiO$_2$ process after 30 min of UV illumination. ([antibiotics]$_0$ = 50 µg/L, [TiO$_2$]$_0$ = 0.1 g/L, pH = 5.8.)

*3.3. Factors Affecting the Efficiency of UV/TiO$_2$ Photocatalysis of Antibiotics*

The pH, TiO$_2$ loadings, and antibiotics concentrations have been shown to be among the most important variables during TiO$_2$ photocatalysis [64]. In the succeeding sections, the effects of initial concentrations of antibiotics, solution pHs and catalyst loadings on TiO$_2$ photocatalysis will be discussed.

3.3.1. Effects of Initial Concentration of Antibiotics

The degradation of antibiotics in livestock wastewater by UV/TiO$_2$ process was carried out by using different initial concentrations of the antibiotics (i.e., 20, 50 and 100 µg/L) for 30 min under UV illumination, and the results are shown in Figures 5 and 6. As seen in Figure 5, the degradation efficiency of nine antibiotics, i.e., ceftiofur, enrofloxacin, oxytetracycline, penicillin-G, penicillin-V, sulfadiazine, sulfamethazine, sulfathiazole, and tetracycline, was the same when using different concentration levels, i.e., represented by 100% degradation after 30 min of UV illumination. The results showed that the removal efficiency of the remaining seven antibiotics (i.e., except sulfamethoxazole) was dependent on their initial concentrations, i.e., the removal efficiency was inversely proportional to the initial concentration of the antibiotics (Figure 6). This result was consistent with the findings of Athanasios et al., showing that the degradation efficiency of sulfamethoxazole by solar-light/TiO$_2$ photocatalysis decreased with the increase in its initial concentrations [65]. Klauson et al. also found the photocatalytic degradation of amoxicillin reduced from 90% to 30% in 6 h by increasing amoxicillin concentration from 10 to 100 mg/L [66]. A possible reason could be the increased competition of antibiotic molecules with the reactive oxidizing species at the high pollutant concentrations [67]. Furthermore, the high concentration of antibiotics may hinder UV light passage through the reaction solution, thereby decreasing the intensity of UV light for the activation of TiO$_2$ photocatalyst [64]. Nevertheless, for sulfamethoxazole, its photocatalytic degradation increased with an increase in its initial concentration. A previous study by Xekoukoulotakis et al. also found that the reaction rate for photocatalytic degradation of sulfamethoxazole increased with increasing its initial concentration in the range of 2.5 to 20 mg/L, consistent with our result [68].

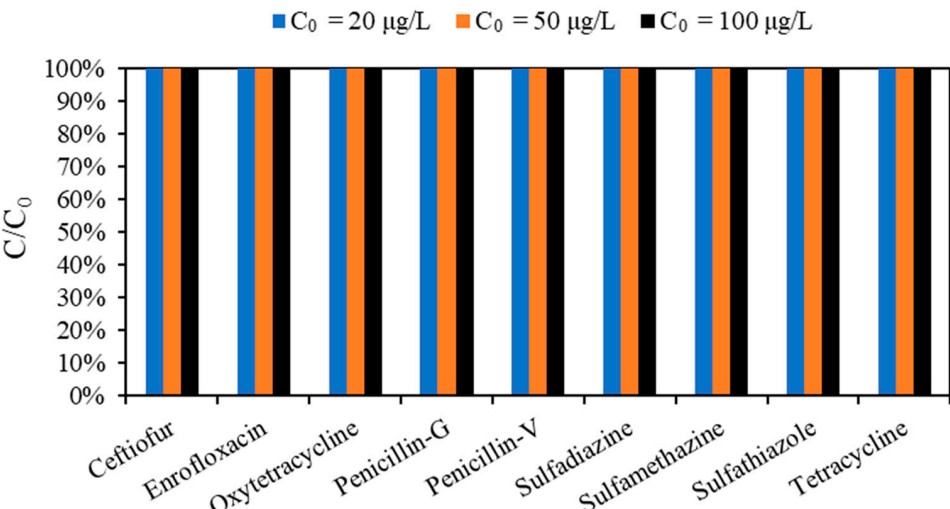

**Figure 5.** Effects of antibiotics concentration using UV/TiO$_2$ photocatalysis for 30 min: Same degradation efficiency at all initial concentration of antibiotics after 30 min illumination.

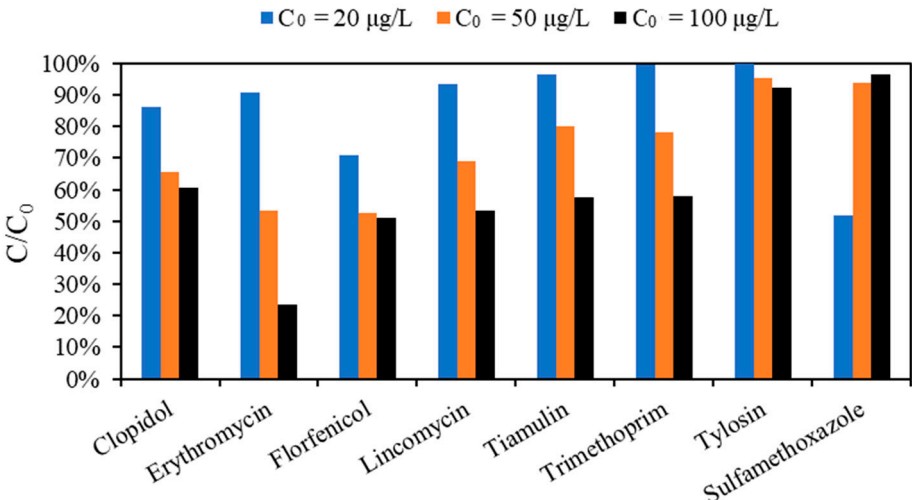

**Figure 6.** Effects of antibiotics concentration using UV/TiO$_2$ photocatalysis for 30 min: High degradation efficiency at low initial concentration of antibiotics.

### 3.3.2. Effects of Solution pHs

To investigate the antibiotic removal efficiency at different solution pHs, three pHs of 3, 5.8, and 9.9 were tested. During the experiment, the photocatalyst concentration was fixed at 0.1 g/L to minimize antibiotic adsorption by the photocatalyst, and the total concentration of antibiotics was 50 µg/L. Figures 7 and 8 show the removal of each antibiotic at different solution pHs after 30 min of UV irradiation. As seen in Figure 7, ten antibiotics, i.e., ceftiofur, oxytetracycline, penicillin-G, penicillin-V, sulfadiazine, sulfamethazine, sulfamethoxazole, sulfathiazole, tetracycline, and tylosin were almost completely decomposed at all the studied pHs. However, other antibiotics have their preference of solution pHs for photocatalysis. As can be seen in Figure 8, clopidol, enrofloxacin, erythromycin, and florfenicol demonstrated over 90% removal efficiency under basic conditions, enrofloxacin showed over 90% removal efficiency at pH 5.8, and lincomycin and tiamulin showed the highest removal efficiency at pH 3. In the case of trimethoprim, the highest removal efficiency was achieved at pH 5.8. Furthermore, in the case of trimethoprim, a pKa value of trimethoprim and TiO$_2$ was 7.2 and 6.7, respectively; these have similar charges, and no electrostatic interactions occurred [69]. This explained why trimethoprim showed a higher removal efficiency at pH 5.8 rather than at pH 3 or 9.9.

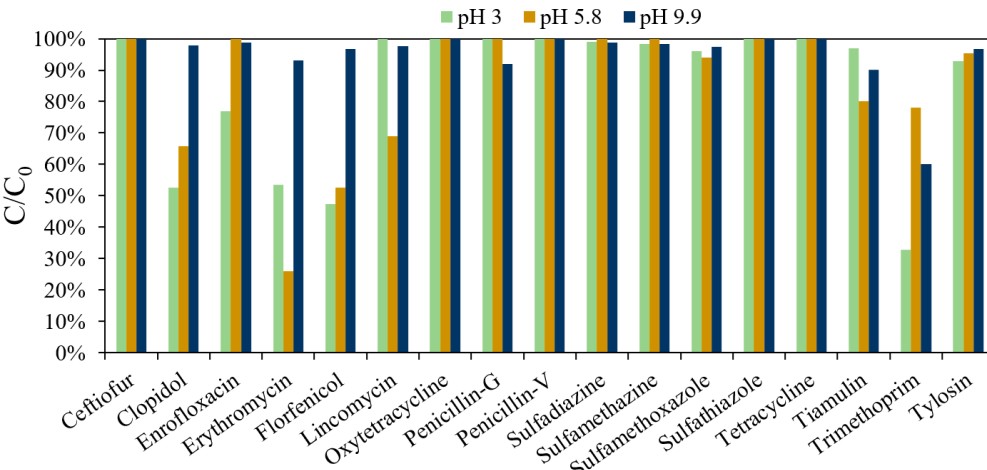

**Figure 7.** Degradation efficiency of antibiotics at different solution pH. [antibiotics]$_0$ = 50 μg/L, [TiO$_2$]$_0$ = 0.1 g/L.

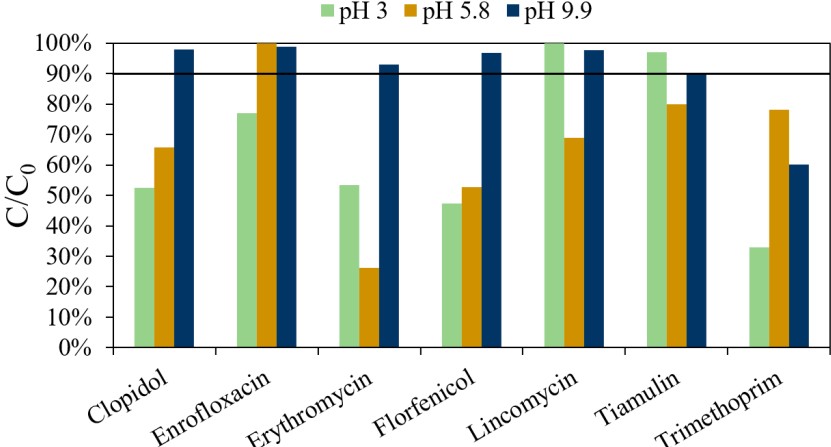

**Figure 8.** The pH value for 90% degradation efficiency of selected antibiotic by using UV/TiO$_2$. [antibiotics]$_0$ = 50 μg/L, [TiO$_2$]$_0$ = 0.1 g/L.

3.3.3. Effects of Photocatalyst Loadings

The degradation of antibiotics in livestock wastewater by UV/TiO$_2$ process was carried out at different TiO$_2$ loadings (i.e., 0.1, 0.3, 0.5, and 1.0 g/L) for 60 min of UV illumination, and the results were shown in Figures 9–13. All the seventeen antibiotics were spiked in the reactor to achieve a total concentration of 350 μg/L. The results showed 34, 40, 40 and 41% degradation of the antibiotic using TiO$_2$ loadings of 0.1, 0.3, 0.5, and 1.0 g/L, respectively, during 5 min UV illumination (Figure 9). After 60 min illumination, more than 90% of the antibiotics were degraded with all TiO$_2$ loadings (Figure 10). The highest degradation efficiency was obtained using 1.0 g/L. Meanwhile, the degradation efficiency of the individual antibiotics under UV/TiO$_2$ process was investigated by using different TiO$_2$ photocatalyst loadings, and the results were shown in Figures 11 and 12. As shown in Figure 11, six antibiotics, i.e., ceftiofur, oxytetracycline, penicillin-V, sulfadiazine, sulfathiazole, and tetracycline, had an insignificant effect by TiO$_2$ loadings, and these antibiotics were completely degraded during 60 min illumination regardless of the photocatalyst loadings. It seemed that these antibiotics might have weak bonds prone to relatively easier decomposition, probably due to effective UV photolysis even though TiO$_2$ loading was low, i.e., 0.1 g/L.

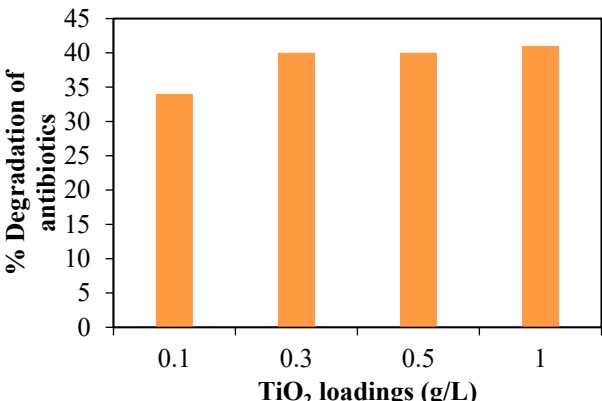

**Figure 9.** The effect of TiO$_2$ loadings on the degradation of total antibiotics after 5 min UV illumination at different TiO$_2$ loadings.

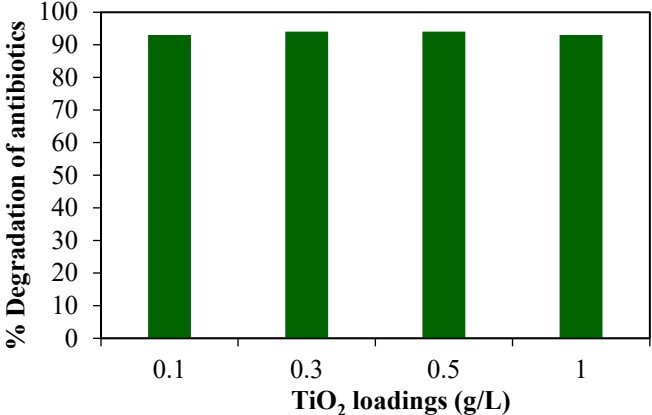

**Figure 10.** The effect of TiO$_2$ loadings on the degradation of total antibiotics after 60 min UV illumination at different TiO$_2$ loadings.

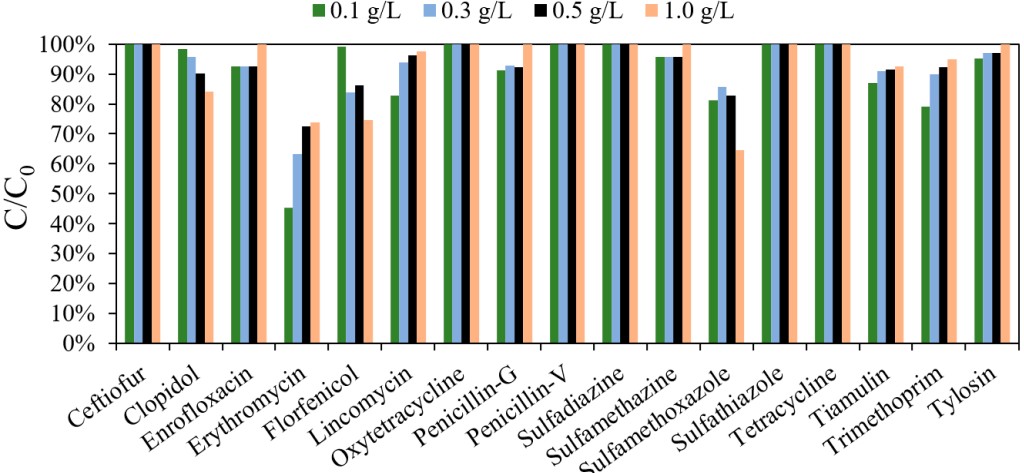

**Figure 11.** Degradation of target antibiotics using UV/TiO$_2$ photocatalysis for 60 min at different TiO$_2$ loadings.

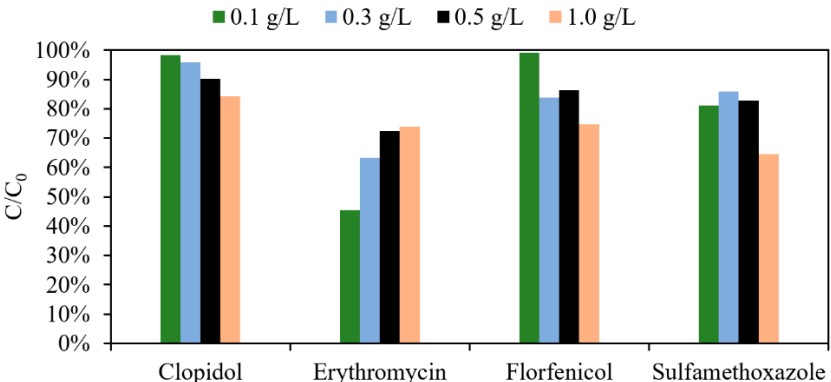

**Figure 12.** Degradation of selected antibiotics with varied degradation efficiencies at different photocatalyst loadings using $UV/TiO_2$ photocatalysis for 60 min.

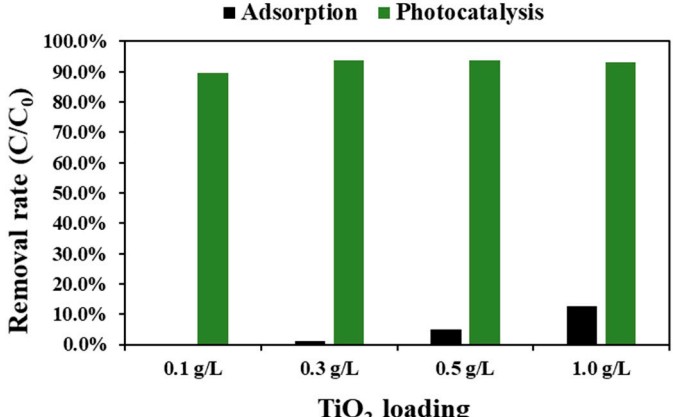

**Figure 13.** Removal rate of antibiotics of adsorption and photocatalysis according to $TiO_2$ loading.

Figure 11 shows that the degradation efficiency of enrofloxacin was 93% at 0.1 to 0.5 g/L, penicillin-G was 91, 93, and 92% at 0.1, 0.3, and 0.5 g/L, respectively, and sulfamethazine was 96% at 0.1 to 0.5 g/L $TiO_2$ loading, which reached 100% at 1.0 g/L $TiO_2$ loading, in all cases. The degradation efficiency of four antibiotics, i.e., lincomycin, tiamulin, trimethoprim, and tylosin, was proportional to $TiO_2$ loading and reached more than 90% degradation at 1.0 g/L. In the case of lincomycin, tiamulin, and trimethoprim, the degradation efficiency was around 80% using a lower $TiO_2$ loading of 0.1 g/L.

However, the degradation of four antibiotics, i.e., clopidol, erythromycin, florfenicol, and sulfamethoxazole, was very different from the other antibiotics, since they had a significant effect by $TiO_2$ loadings, as shown in Figure 12. They were not completely degraded by $UV/TiO_2$ photocatalysis, even though the $TiO_2$ loading increased up to 1.0 g/L.

Figure 13 shows the adsorption capacity of the photocatalyst when using different loadings. As the photocatalyst concentration increased from 0.1 to 1.0 g/L, the adsorption efficiency of the antibiotics increased from 0.1 to 1.2, 4.8, and 12.6%, respectively. In the case of $TiO_2$ photocatalysis (Figure 13), as the catalyst concentration increased from 0.1 to 1.0 g/L, the degradation efficiency was also increased from 89.5 to 93.2%. For improving the overall efficiency of antibiotics removal, adsorption is an important parameter. Elmolla and Chaudhuri [51] found the removal efficiency of amoxicillin, ampicillin and cloxacillin by $UV/TiO_2$ photocatalysis after 300 min irradiation increased from 42.3 to 58.7, 33.3 to 52.4 and 46.6 to 60.2, by increasing $TiO_2$ concentrations from 0.5 to 2.0 g/L, respectively.

3.3.4. $UV_{254}$ Reduction in the Wastewater Samples

The reduction in UV 254 nm absorption by antibiotics in the wastewater samples was measured [70] to confirm the degradation of the antibiotics, which have aromatic

contents. According to the Lambert–Beer law [71], light absorbance is directly proportional to pollutants and dissolved natural organic matter concentrations in samples. Using this law, the changes in concentrations of the antibiotics and dissolved natural organic matter were monitored. As can be seen in Figure 14, concentration ($C/C_0$) dramatically reduced when samples were treated with UV/TiO$_2$ photocatalysis. However, there was a plateau of the degradation of dissolved organic compounds by photocatalysis. The reduction in UV$_{254}$ was associated with the degradation of aromatic compounds and the consequent transformation to smaller molecules, e.g., formic or acetic acids, aldehydes, ketone etc. [52]. The results showed that the degradation of antibiotics by the UV/TiO$_2$ process did not result in full mineralization of antibiotics, probably due to the complex structures and strong bonds of the antibiotics. It might be due to competition among antibiotics and dissolved natural organic matter for the reactive species, during UV/TiO$_2$ photocatalysis. Further mineralization of the antibiotics was studied from the COD and TOC removal efficiency in the next section.

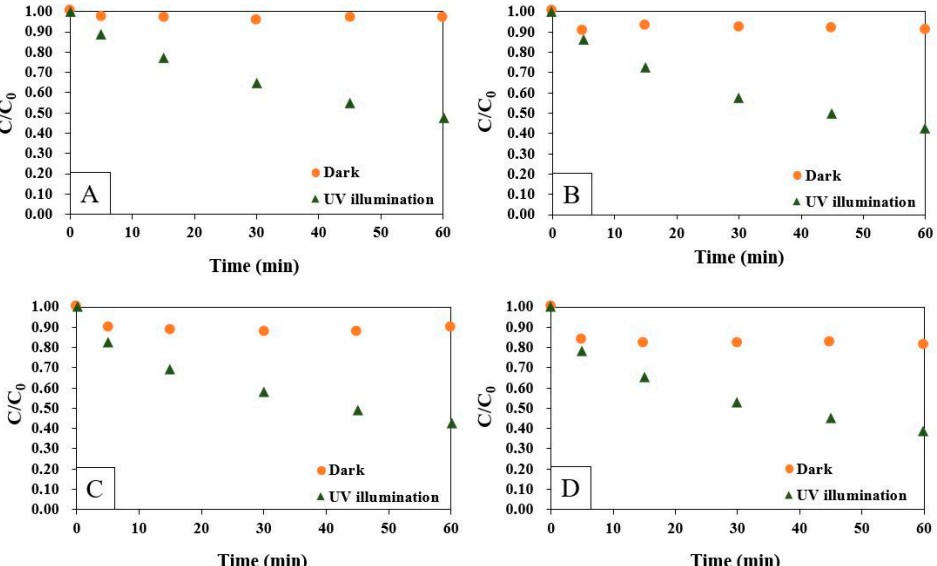

**Figure 14.** Changes of UV absorption by total dissolved organic compounds during UV/TiO$_2$ photocatalysis at different TiO$_2$ loadings: (**A**) 0.1 g/L, (**B**) 0.3 g/L, (**C**) 0.5 g/L, and (**D**) 1.0 g/L.

*3.4. Mineralization Studies*

The mineralization of toxic organic molecules into innocent inorganic compounds, such as H$_2$O and CO$_2$, was studied by using different techniques. The mineralization of the antibiotics was measured from COD and TOC analysis, and the results are shown in Figure 15. The results showed that 13% COD and 10% TOC removal was achieved in 60 min at photocatalyst concentrations of 1.0 g/L. The results showed that the mineralization efficiency (i.e., COD and TOC removal) was far below the degradation efficiency of antibiotics, consistent with the literature studies [64]. A possible reason was the generation of smaller molecules and reaction intermediates during the degradation process, which need extra $^\bullet$OH for their degradation [64]. Elmolla and Chaudhuri [52] found the degradation efficiency of antibiotics in water by UV/TiO$_2$ photocatalysis was 59% in 300 min, while the corresponding mineralization efficiency was only 10%, attributed to the same reasons.

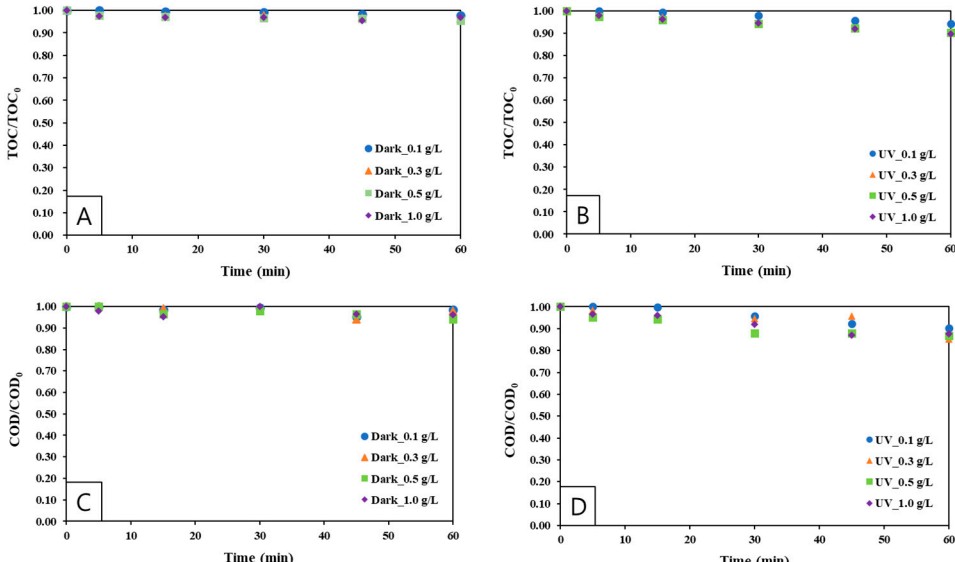

**Figure 15.** TOC removal under (**A**) dark conditions and (**B**) UV illumination, and COD removal under (**C**) dark conditions and (**D**) UV illumination.

Figure 15 shows the removal of TOC and COD under dark conditions and UV conditions. TOC removal was observed to be only 2, 4, 4, and 3% as photocatalyst concentrations of 0.1, 0.3, 0.5, and 1.0 g/L, respectively, under dark conditions (Figure 15A). However, under UV illumination, the TOC removal increased up to 10% at 1.0 g/L of $TiO_2$ (Figure 15B).

As shown in Figure 15C,D, the average COD removal was 3% (minimum 1% and maximum 6%) in dark conditions. However, the average COD removal reached 13% (minimum 10% and maximum 15%) when under UV illumination. The trend of COD removal was very similar to TOC removal, indicating the antibiotic could not be fully decomposed by $TiO_2$ photocatalysis under these experimental conditions. However, Xekoukoulotakis et al. [68] and Abellán et al. [59] observed a nearly complete mineralization of sulfamethoxazole using UVA/$TiO_2$ photocatalysis.

### 3.5. Comparison of Different AOPs

The degradation of the seventeen antibiotics was carried out for 30 min under illumination ([antibiotics]$_0$ = 20 μg/L) under the following conditions: (i) UV only, (ii) photocatalyst only and (iii) UV/photocatalyst, and the results were shown in Figure 16. As seen, the degradation efficiency of the antibiotics was followed as: UV/photocatalyst (photocatalysis) > Only UV (photolysis) > only photocatalyst (adsorption) (Figure 16). The results were consistent with the literature studies, showing higher degradation efficiency of meropenem antibiotics in water by UV/photocatalyst systems rather than UV photolysis [72]. Shankaraiah et al. [73] reported 45 and 90% degradation of antibiotic norfloxacin in aqueous solution by UV and UV/$TiO_2$ processes, respectively. The results were attributed to the generation of •OH by the UV/$TiO_2$ photocatalyst system (reactions 1–4), which can attack most of the organic molecules in an aqueous solution [74]. On the other hand, the degradation of the antibiotics by only the UV system was attributed to the UV absorbance and photolysis of the double bond contained in the molecules (i.e., antibiotics) [60]. The usually high removal efficiency of antibiotics by UV/$TiO_2$ process rather than only UV might suggest that the studied antibiotics had a strong affinity towards oxidation by •OH.

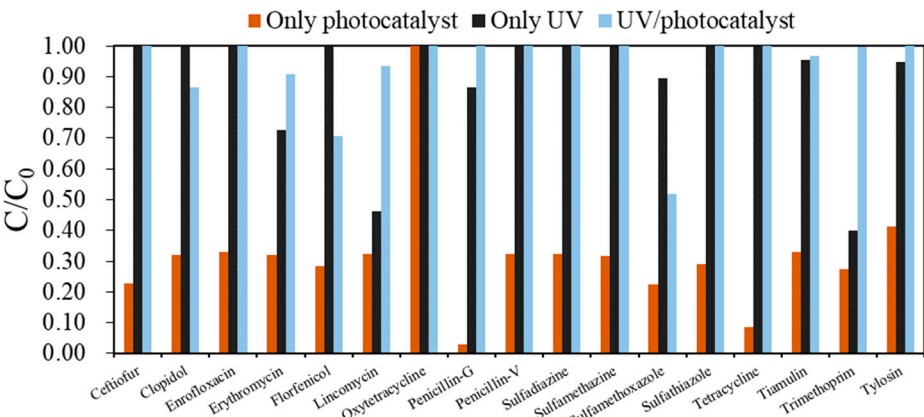

**Figure 16.** Antibiotic decomposition by using different AOPs, i.e., (i) only UV (ii) only $TiO_2$ photocatalyst, and UV/$TiO_2$ photocatalyst.

However, in the case of clopidol, florfenicol, and sulfamethoxazole, the antibiotic degradation efficiency by the only UV system was superior to the UV/photocatalyst or only photocatalyst systems (Figure 17). This result could be attributed to any high UV photon absorption capacity for those antibiotics, whereby the UV photon intensity may decrease in the presence of photocatalyst, thereby retarding the degradation efficiency. Despite the fact that the antibiotics were significantly decomposed even under only the UV conditions, the UV/$TiO_2$ process was more effective in removing all the antibiotics contained in the livestock wastewater effluent. Several research studies reported low removal efficiency of antibiotics by UV, Visible or solar photolysis [63,75]. Elmolla and Chaudhuri [63] reported that UV photolysis resulted in 2.9, 3.8, and 4.9% removal of amoxicillin, ampicillin, and cloxacillin in an aqueous solution, respectively, under 300 min illuminations. Che et al. found that ciprofloxacin (CIP) and tetracycline (TC) were resistant towards the visible light irradiation, represented by less than 5% degradation in 2 h illumination [76]. Meanwhile, Priyaa et al. reported that ampicillin (AMP) and oxytetracycline (OTC) antibiotics were quite stable under solar light irradiation, represented by less than 5% degradation in 2 h illumination [75].

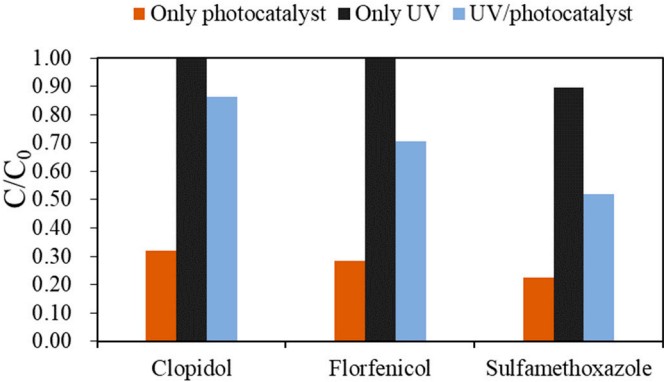

**Figure 17.** Exceptional behavior of the specific antibiotic towards (i) only UV (ii) only $TiO_2$ photocatalyst and UV/$TiO_2$ photocatalyst.

## 4. Conclusions

The UV/$TiO_2$ photocatalysis was used for the treatment of real livestock wastewater effluents. The degradation efficiency of antibiotics was only slightly affected by the photocatalyst loadings at the experimental conditions. However, the removal efficiency was inversely proportional to the initial concentration of antibiotics. In the case of solution pHs, due to different speciation and UV absorption of antibiotics, there is a pH preference

for each antibiotic. However, a high removal efficiency was observed for many antibiotics regardless of solution pHs.

The degradation efficiency of antibiotics by UV/TiO$_2$ photocatalysis was very high, i.e., >90% after 60 min illumination. However, the extent of mineralization was significantly low, as confirmed by the removal of COD and TOC or the reduction in UV$_{254}$ absorbance. The overall degradation efficiency of the antibiotics followed the order: UV/TiO$_2$ photocatalyst > Only UV > only TiO$_2$ photocatalyst systems. However, it was revealed that low antibiotic concentrations of some studies were completely decomposed by only UV photolysis as well.

Nevertheless, the AOPs may have a disadvantage of high cost in real-life applications, which can be largely minimized by the activation of photocatalyst by sustainable solar light energy, accomplishable through narrowing the band-gape energy by doping of photocatalyst, and it can be investigated in our future studies.

The separation of photocatalyst after use, particularly regeneration of the photocatalyst for reuse, is a desirable quality for practical uses. The suspended TiO$_2$ particles can be separated and/or recovered by filtration, followed by regeneration via calcination. Alternatively, the replacement of powdered TiO$_2$ photocatalysts with fixed-bed continuous columns, or especially with catalyst immobilization on supporting materials (e.g., TiO$_2$ thin film on glass plates, beads etc.), is a more convenient approach compared to other methods, which needs to be investigated in our future studies. Samuel Moles et al. reported reprocessing completely, or an 80% removal of antibiotics from wastewater from pilot-scale plants, and the catalyst was recovered and reused after retreatment [77].

The application of UV/TiO$_2$ system for the simultaneous removal of a vast number of different antibiotics from real livestock wastewater effluents (i.e., one and a half dozen) is a significant contribution towards the real-life application. However, this research is a lab-scale test, and thus, a prototype or plant-scale test should be conducted to industrialize this technology in the livestock wastewater treatment facilities. It was concluded that UV/TiO$_2$ photocatalysis was a promising technology for the treatment of real livestock wastewater effluents.

**Author Contributions:** Investigation, Y.P., S.K. (Sanghyeon Kim) and J.K.; Writing-original draft preparation, Y.P.; Writing-review and editing, S.K. (Sanghyeon Kim), J.K., S.K. (Sanaullah Khan), C.H.; Experimentation, Y.P.; Supervision, C.H.; Project administration, C.H.; Funding acquisition: C.H. All authors have read and agreed to the published version of the manuscript.

**Funding:** National Research Foundation of Korea (NRF) grant funded by the Korean government (MSIT) (No. 2021R1A2C1093183) and INHA University Research Grant (INHA-62979-1).

**Institutional Review Board Statement:** Not applicable.

**Informed Consent Statement:** Not applicable.

**Acknowledgments:** CH acknowledges the support of the National Research Foundation of Korea (NRF) grant funded by the Korean government (MSIT) (No. 2021R1A2C1093183) and INHA University Research Grant (INHA-62979-1).

**Conflicts of Interest:** The authors declare no conflict of interest.

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
