# Peer review of "UV/TiO2 Photocatalysis as an Efficient Livestock Wastewater Quaternary Treatment for Antibiotics Removal"

_water, doi:10.3390/w14060958_

Round 1

Reviewer 1 Report

The present manuscript described the elimination of 17 antibiotics from real livestock wastewater effluents by an UV/TiO2 advanced oxidation process.

I believe the manuscript matched the scientific scope of Water, and can be considered for publication after minor revisions, especially related to expression in English, some of which are listed below:

- In the title,  the word ”quaternary” is not justified so, must be removed.

 -At pag. 2 line 52  phrase ”It was reported that the bacteria generated in antibiotic-resistant environments, i.e., antibiotics resistant bacteria, can directly or indirectly enter the human body [18]” must be changed with ”Photocatalyst resistant bacteria, may introduce into the environment, which can directly or indirectly invade the human body [18]”.

-At pag. 2, line 61,” for completely removing” must be changed with ”to remove completely”.

-At pag. 2, line 68,” sludge produced” must be changed with ”produced sludge”.

-At pag. 2, line 85,” iron is catalyzed” must be changed with ” iron is the catalyst”.

-At pag. 2, line 91,” hydroxyl radicals” must be changed with ” different reactive oxygen species”.

-At pag. 3, line 104,” was measured” must be changed with ” was determined”.

-At pag. 3, line 115,” was provided” must be changed with ” is presented”.

-At pag. 3, line 118 phrase “The TiO2 used as a photocatalyst was Aeroxide P25, and the image was shown in Figure 1.” must be changed with “The TiO2 used as a photocatalyst was Aeroxide P25”. Phrase ” Morphological characterization of the photocatalyst by TEM and SEM microscopies is presented in Figure 1” must be introduced at 3.1. Characteristics of TiO2.

- In Table 1 the characteristic parameters of the real livestock wasterwater must be specified by the complete name and not only by initials.

- At pag. 7, line 161,” was shown” must be changed with ” is shown”.

- At pag. 7, line 163,” The pairs of e− and h+ generated to CB and VB, respectably, produce •OH” must be changed with ” The pairs of e− and h+ generated in CB and VB, respectively , produce oxygen reactive species  •OH and O2−• ”. In Figure 3 OH+ must be replaced by OH-.

- At pag. 7, line 169, pag. 10, line 249” were shown” must be changed with ” are shown”.

- At pag. 8, line 201, pag. 10, line 248 ” at using” must be changed with ” by using”.

- At pag. 8, line 202,” were shown” must be changed with ” are shown”.

- At pag. 8, line 205,” at using” must be changed with ” when using”.

- At pag. 9, line 229,” the concentration of the total antibiotics” must be changed with ” the total concentration of antibiotics”.

-At pag. 9, line 234,” As seen” must be changed with ” As can be seen”.

-At pag. 9, line 240,” and do not react to electrostatic interactions” must be changed with ” no electrostatic interactions occurred”.

- Legend of Figure 7 ”The pH preference of selected antibiotics with about 90% using UV/TiO2” must be changed with ” The pH values for 90% degradation  efficiency of selected antibiotic by using UV/TiO2 treatment”

- For subchapter 3.3.3. Effects of photocatalyst loadings, pag. 10, all the text must be reformulated; The TiO2 loading has an influence on degradation of tested antibiotics and not vice versa.

- Title of ”3.3.4. UV254 reduction by the antibiotics samples” must be changed to UV254 reduction in the wastewater samples and the text must be revised

--At pag. 14, line 314,” Mineralization is the conversion of toxic organic molecules into innocent inorganic compounds, such as H2O and CO2 by using different techniques.” must be changed with ” Mineralization of toxic organic molecules into harmless inorganic compounds, such as H2O and CO2  was studied by using different techniques.”

Finally I recommend that the entire manuscript should be checked for English expression.

Author Response

The present manuscript described the elimination of 17 antibiotics from real livestock wastewater effluents by an UV/TiO2 advanced oxidation process.

I believe the manuscript matched the scientific scope of Water, and can be considered for publication after minor revisions, especially related to expression in English, some of which are listed below:

- In the title, the word ”quaternary” is not justified so, must be removed.

Answer: Thank you very much for the comment. The term of “Quaternary Wastewater” has been widely used to remove micropollutants in wastewater and thus, we decided to keep it in the title.

 -At pag. 2 line 52  phrase ”It was reported that the bacteria generated in antibiotic-resistant environments, i.e., antibiotics resistant bacteria, can directly or indirectly enter the human body [18]” must be changed with ”Photocatalyst resistant bacteria, may introduce into the environment, which can directly or indirectly invade the human body [18]”.

Answer: Thank you very much for the comment. It was corrected.

-At pag. 2, line 61,” for completely removing” must be changed with ”to remove completely”.

Answer: Thank you very much for the comment. It was corrected.

-At pag. 2, line 68,” sludge produced” must be changed with ”produced sludge”.

Answer: Thank you very much for the comment. It was corrected.

-At pag. 2, line 85,” iron is catalyzed” must be changed with ” iron is the catalyst”.

Answer: Thank you very much for the comment. It was corrected.

-At pag. 2, line 91,” hydroxyl radicals” must be changed with ” different reactive oxygen species”.

Answer: Thank you very much for the comment. It was corrected.

-At pag. 3, line 104,” was measured” must be changed with ” was determined”.

Answer: Thank you very much for the comment. It was corrected.

-At pag. 3, line 115,” was provided” must be changed with ” is presented”.

Answer: Thank you very much for the comment. It was corrected.

-At pag. 3, line 118 phrase “The TiO2 used as a photocatalyst was Aeroxide P25, and the image was shown in Figure 1.” must be changed with “The TiO2 used as a photocatalyst was Aeroxide P25”. Phrase ” Morphological characterization of the photocatalyst by TEM and SEM microscopies is presented in Figure 1” must be introduced at 3.1. Characteristics of TiO2.

Answer: Thank you very much for the comment. It was corrected. Also, we relocated TEM, SEM, XRD, and FTIR analyses in the section 3.1.

- In Table 1 the characteristic parameters of the real livestock wasterwater must be specified by the complete name and not only by initials.

Answer: Thank you very much for the comment. It was corrected.

- At pag. 7, line 161,” was shown” must be changed with ” is shown”.

Answer: Thank you very much for the comment. It was corrected.

- At pag. 7, line 163,” The pairs of e− and h+ generated to CB and VB, respectably, produce •OH” must be changed with ” The pairs of e− and h+ generated in CB and VB, respectively , produce oxygen reactive species  •OH and O2−• ”. In Figure 3 OHmust be replaced by OH-.

Answer: Thank you very much for the comment. It was corrected.

- At pag. 7, line 169, pag. 10, line 249” were shown” must be changed with ” are shown”.

Answer: Thank you very much for the comment. It was corrected.

- At pag. 8, line 201, pag. 10, line 248 ” at using” must be changed with ” by using”.

Answer: Thank you very much for the comment. It was corrected.

- At pag. 8, line 202,” were shown” must be changed with ” are shown”.

Answer: Thank you very much for the comment. It was corrected.

- At pag. 8, line 205,” at using” must be changed with ” when using”.

Answer: Thank you very much for the comment. It was corrected.

- At pag. 9, line 229,” the concentration of the total antibiotics” must be changed with ” the total concentration of antibiotics”.

Answer: Thank you very much for the comment. It was corrected.

-At pag. 9, line 234,” As seen” must be changed with ” As can be seen”.

Answer: Thank you very much for the comment. It was corrected.

-At pag. 9, line 240,” and do not react to electrostatic interactions” must be changed with ” no electrostatic interactions occurred”.

Answer: Thank you very much for the comment. It was corrected.

- Legend of Figure 7 ”The pH preference of selected antibiotics with about 90% using UV/TiO2” must be changed with ” The pH values for 90% degradation  efficiency of selected antibiotic by using UV/TiO2 treatment”

Answer: Thank you very much for the comment. It was corrected.

- For subchapter 3.3.3. Effects of photocatalyst loadings, pag. 10, all the text must be reformulated; The TiO2 loading has an influence on degradation of tested antibiotics and not vice versa.

Answer: Thank you very much for the comment. As commented, in general, the efficiency of photocatalysis increased with high TiO2 loadings and this study also shows the increased efficiency of the photocatalysis. However, in some cases of different antibiotics of Cloidol, Flofernicol, and Sulfamethoxazole, their degradation was different than the common phenomena due to the competition among different antibiotics.

We appropriately revised the section carefully based on the comment.

- Title of ”3.3.4. UV254 reduction by the antibiotics samples” must be changed to UV254 reduction in the wastewater samples and the text must be revised

Answer: Thank you very much for the comment. It was corrected.

--At pag. 14, line 314,” Mineralization is the conversion of toxic organic molecules into innocent inorganic compounds, such as H2O and CO2 by using different techniques.” must be changed with ” Mineralization of toxic organic molecules into harmless inorganic compounds, such as H2O and CO2  was studied by using different techniques.”

Answer: Thank you very much for the comment. It was corrected.

Finally I recommend that the entire manuscript should be checked for English expression.

Answer: Thank you very much for the comment. We extensively checked the whole manuscript to significantly improve the level of English with all coauthors.

Reviewer 2 Report

The authors proposed TiO2 photocatalyst for the photodegradation of antibiotics. The removal of this type of pollutants must be investigated due to its negative impacts on human health and environment as well. The information is presented in such a way that scientists can understand it. However, the characterization aspect of photocatalyst was insufficient. The practical aspect was reasonably studied. This manuscript needs minor revision and I have listed these issues and recommendations in chronological order. Following is a summary of the minor corrections and revisions:

1.The authors should enhance the novelty and importance of this work in the introduction section.

2.The characterization of TiO2 photocatalyst was poor. The authors should add more characterization (e.g. FTIR and XRD analysis).

3.In order to make the adsorption process more feasible, the adsorbent is usually regenerated. Could the authors explain possible ways to perform the regeneration of the photocatalyst. This information should be mentioned for future studies.

4.The major drawback of this paper is followed by several questions: Can this work be feasible to be done in industrial scale, and can it be scaled up? What is the novelty of this work in context of related composites used for the removal of other similar effluents compositions?

5.Can the same experiments be done using continuous adsorption column? Can this method be scaled up? Cite updated papers in the said query, include it in the introduction, and conclusion part of your revised Manuscript.

6.Conclusions need to be improved by specifying the discussed important points within this work. In the conclusions, the authors should also provide an outlook of the challenges and potential future directions.

Reviewer 3 Report

Removal of antibiotics and their residuals from water has high relevancy for the science and for the practice, as well. Advanced oxidation processes are considered as promising technology for the removal of pharmaceutical compounds from water. UV/TiO2 photocatalytic has theoretically high efficiency but detailed analysis is needed to obtain the real degradation efficiency and kinetics, investigate the matrix effects of other pollutants and components, and to investigate the effects of process parameters and optimize them. Therefore, the topic of the manuscript can be considered as interesting for the readers. The manuscript is generally well written with a logic structure. Introduction section summarizes well the background of the research, the relevance and importance of antibiotics removal, the applicability of AOPs with special focus on TiO2 photocatalysis, and the specific research motivations. The applied methods (photocatalyst characterization, analytical methods) are adequate to the sample characteristics. Materials and methods are described clearly. The manuscript contains interesting and valuable results not just for the science but also for the practice. Effects of process parameters (antibiotic concentration, pH, photocatalyis loading), and mineralization of toxic organic molecules are presented well and clear and discussed in details with relevant references. Comparison of the different removal methods are presented clearly, as well.

Comments, suggestions:

I suggest the authors to give clearly the novelties of the study (Introduction section).

It is not clear how was selected the concentration range of different antibiotics for the degradation/oxidation tests?

I suggest the authors to discuss briefly the scale-up of the process (problems, possibilities), and the economy and/or energy efficiency of the process (related to UV generation, for instance).

Which separation methods are suggested for the catalysts recycling (for the practice)?
